# Sex-specific tuning of modular muscle activation patterns for locomotion in young and older adults

**Alessandro Santuz**[1,2]*, **Lars Janshen**[1,2], **Leon Brüll**[1,2,3], **Victor Munoz-Martel**[1,2],
**Juri Taborri**[4], **Stefano Rossi**[4], **Adamantios Arampatzis**[1,2]

**1** Department of Training and Movement Sciences, Humboldt-Universität zu Berlin, Berlin, Germany,
**2** Berlin School of Movement Science, Humboldt-Universität zu Berlin, Berlin, Germany, **3** Network Aging
Research, Heidelberg University, Heidelberg, Germany, **4** Department of Economics, Engineering, Society
and Business Organization, University of Tuscia, Viterbo, Italy

* alessandro.santuz@hu-berlin.de

Innsbruck, AUSTRIA

**Data Availability Statement:** All the recordings
and the code used for the analysis can be
downloaded from the supplementary data set,

## Abstract

There is increasing evidence that including sex as a biological variable is of crucial impor-
tance to promote rigorous, repeatable and reproducible science. In spite of this, the body of
literature that accounts for the sex of participants in human locomotion studies is small and
often produces controversial results. Here, we investigated the modular organization of
muscle activation patterns for human locomotion using the concept of muscle synergies
with a double purpose: i) uncover possible sex-specific characteristics of motor control and
ii) assess whether these are maintained in older age. We recorded electromyographic activi-
ties from 13 ipsilateral muscles of the lower limb in young and older adults of both sexes
walking (young and old) and running (young) on a treadmill. The data set obtained from the
215 participants was elaborated through non-negative matrix factorization to extract the
time-independent (i.e., motor modules) and time-dependent (i.e., motor primitives) coeffi-
cients of muscle synergies. We found sparse sex-specific modulations of motor control.
Motor modules showed a different contribution of hip extensors, knee extensors and foot
dorsiflexors in various synergies. Motor primitives were wider (i.e., lasted longer) in males in
the propulsion synergy for walking (but only in young and not in older adults) and in the
weight acceptance synergy for running. Moreover, the complexity of motor primitives was
similar in younger adults of both sexes, but lower in older females as compared to older
males. In essence, our results revealed the existence of small but defined sex-specific differ-
ences in the way humans control locomotion and that these are not entirely maintained in
older age.

## Introduction

For many decades, the vast majority of basic and clinical research studies using non-human
animals failed to either include females or to consider sex as a biological variable, often without
scientifically funded reason [1–3]. In the last few years, and especially in human research, this

accessible at Zenodo (doi:10.5281/zenodo.5171754).

**Funding:** The article processing charge was funded by the Deutsche Forschungsgemeinschaft (DFG, German Research Foundation) - 491192747 and the Open Access Publication Fund of Humboldt-Universität zu Berlin.

**Competing interests:** The authors have declared that no competing interests exist.

trend started changing [2]. Scientists are becoming more and more aware that the physiological differences between males and females must be accounted for in order to promote rigorous, repeatable and reproducible science [4, 5].

Human locomotion, with its highly stereotyped patterns, has been an ideal subject of neuroscience and biomechanics studies for more than one century [6]. Some sex-specific characteristics, often too subtle to quantify [7, 8], of human walking and running have been unraveled, such as the different hip-, knee- and ankle-joint kinematics or the body height-dependent preferred speed and step frequency [9–11]. Females have been shown to walk with greater hip flexion and lower knee extension at touchdown than males, generating greater mechanical joint power from those two joints during the propulsion phase [9]. Moreover, older females showed smaller range of motion of the hip joint during walking in the sagittal plane, the opposite being true in the frontal plane [10]. In younger participants these findings were reversed [7] and a greater arm swing and pelvic obliquity in the frontal plane was noted in females. In general, it is clear that the quantitative assessment of anecdotal and qualitative observations is difficult to obtain and only few studies endeavored to rigorously quantify the sex specificity of kinematic and kinetic parameters during locomotion [8]. Similarly, only a few studies analyzed the way in which females and males activate muscles to produce locomotion. Some of those works reported no effect of sex on the way muscles are activated during either walking [12–17] or running [18, 19]. Others reported higher or earlier muscle activations in females than in males during either walking [19–22] or running [23–26], with females sometimes showing a higher coactivation of agonist and antagonists muscles during walking [27, 28]. However, all the mentioned studies only considered a limited amount of muscles (from a minimum of one and until a rare maximum of eight), thus limiting further analysis of the coordination between muscle groups during the various phases of the gait cycles. Moreover, it might be relevant to understand if potential sex-specific muscle coordination patterns are maintained in older age, since it is known that the neuromotor strategies used by young people differ from those employed by older adults [16, 22, 29–34].

The purpose of this study was to investigate the muscle activation patterns for locomotion in males and females of different ages in order to: i) find possible sex-specific characteristics of motor control and ii) assess whether sex similarities or differences in movement coordination are maintained in older age. To investigate the first point, we adopted the framework of muscle synergies. Based on the hypothesis that the central nervous system can simplify the production of movement by activating muscles in coordinated groups rather than individually [35], muscle synergies [36] are extracted from electromyographic (EMG) activities. In the past twenty years, a number of scientific works adopted the concept of muscle synergies and their modular organization to better understand the neurophysiological mechanisms underlying the generation of locomotor patterns. From the basic comparison of different locomotion modes [37] to the more elaborate investigations on development [38], aging [33] and pathology [39], muscle synergies are a widespread tool for the assessment of motor coordination during walking and running. Yet, to the best of our knowledge, sex as a biological variable has never been the focus of a scientific work analyzing muscle synergies for locomotion. We recorded the EMG activity of 13 lower limb muscles (right side) in females and males during locomotion and then extracted the time-independent (i.e., motor modules) and time-dependent (i.e., motor primitives) coefficients of muscle synergies through non-negative matrix factorization. To assess the possible influence of aging on sex-specific coordination patterns, we analyzed walking in young and older adults as well as running in young adults. We hypothesized that small, if any, sex-specific modulations would emerge from the analysis of muscle synergies and of gait temporal parameters. Moreover, based on previous findings [30, 31, 33], we hypothesized that any sex-related tuning of muscle synergies might depend on the age group of the participants.

## Results

### Gait cycle temporal parameters

During walking, the duration of the stance phase (Figs 1 and 2A) was moderately affected by both sex (β = -35.57, CrI = [-58.65, -12.33]) and age (β = 15.95, CrI = [-1.04, 32.79]), with an interaction between the two. The *post-hoc* analysis revealed a stronger effect of sex on the stance duration in older adults, while in both age groups females spent less time than males in the stance phase (S1 Fig). It might look like the different walking speeds played an important role; however, this effect was associated with a large estimation uncertainty, as visible from the large width of the credible interval (β = 160.86, CrI = [52.90, 269.44]). Specifically, the walking speed in young adults ranged between 1.1 and 1.5 m/s, while in older adults it spanned between 1.0 and 1.4 m/s (see also Table 1). The swing duration (Figs 1 and 2B) was moderately affected by both sex (β = -16.8, CrI = [-34.01, 0.34]) and age (β = -15.66, CrI = [-28.22, -3-31]), with negligible interaction between the two. Regardless of age, females showed shorter swing

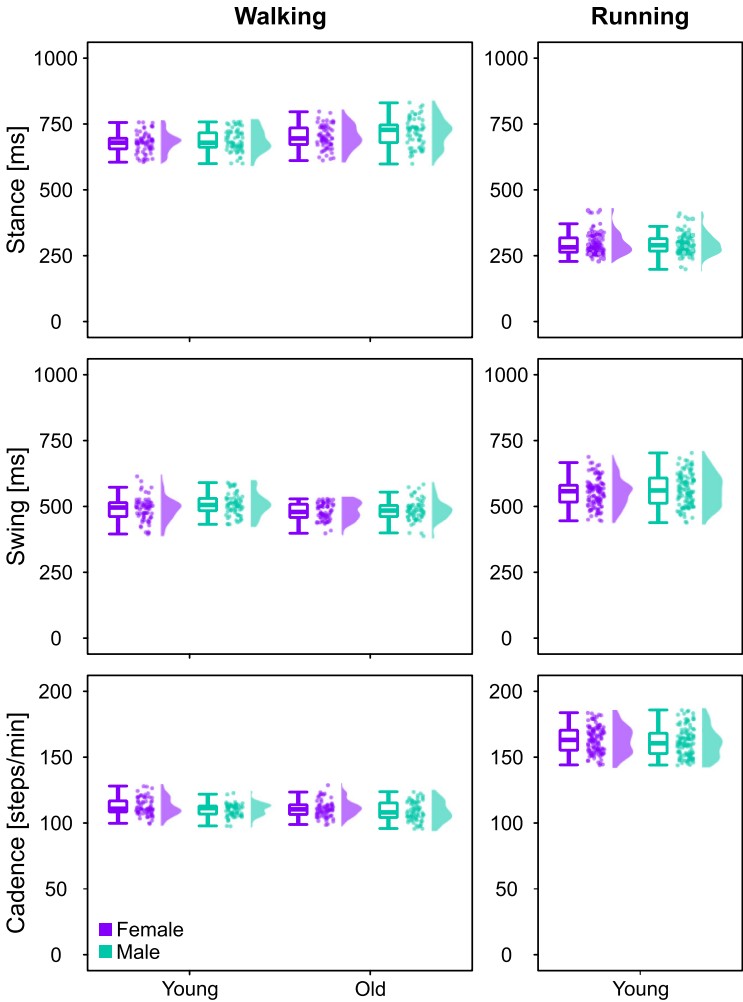

**Fig 1. Gait temporal parameters.** Boxplots describing the three gait temporal parameters stance duration, swing duration and cadence for the three investigated groups, calculated from walking or running. Raw data points (dots represent the mean value of 30 gait cycles from every participant) and their density estimates are presented to the right side of each boxplot.

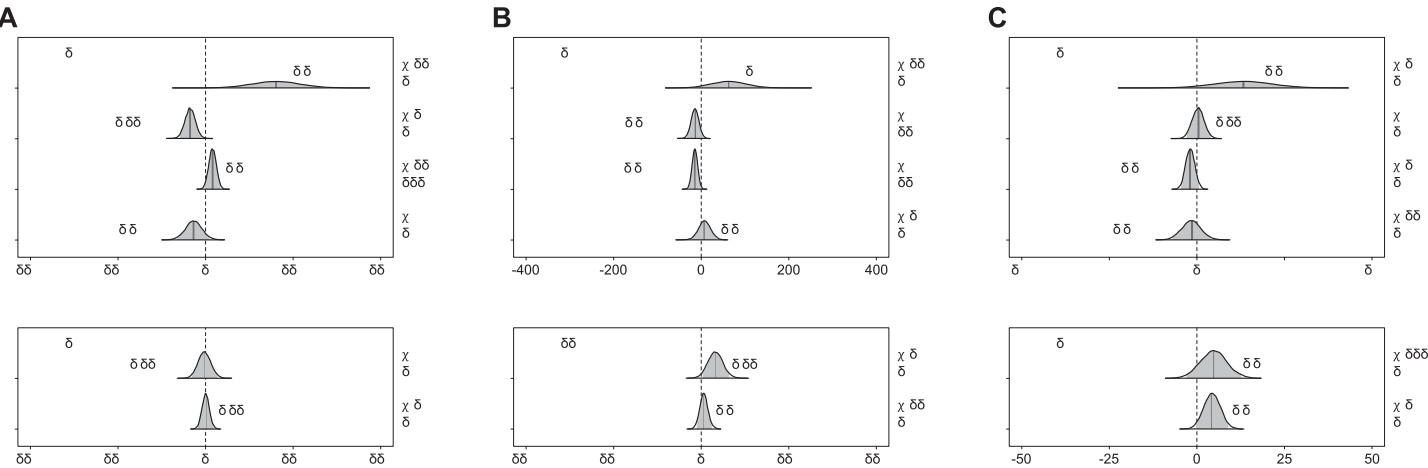

**Fig 2. Posterior 95% credible intervals for the estimated differences in gait temporal parameters.** (A) The 95% credible intervals and their probability distributions (shaded areas) describe the effects and interaction of sex and age (with locomotion speed as a covariate) on the stance duration. Positive values indicate longer stance times in females than in males and in older than in younger adults. Effect size in the style of Hedges are shown as $\delta_t$. A Bayesian equivalent of $R^2$ is presented as $R^2_{Bayes}$. The $\chi^2$ and p-values resulting from a frequentist linear mixed model analysis are reported as additional information. (B) The same as in panel (A), but for the swing phase duration. (C) The same as in panel (A), but for the cadence or number of steps per minute.

duration than males, while older adults had shorter swing times than younger participants. For cadence, or number of steps per minute (Figs 1 and 2C), there was only a small (if any) effect of sex ($\beta$ = 0.44, CrI = [-2.98, 3.92]) and age ($\beta$ = -1.98, CrI = [-4.52, 0.61]). The considerations about the effect of walking speed are also valid for the swing times ($\beta$ = 62.94, CrI = [-14.78, 142.34]) and cadence ($\beta$ = 13.21, CrI = [-3.30, 29.41]).

In running, sex had a negligible effect on the stance times ($\beta$ = 1.48, CrI = [-15.22, 18.76]) and the different locomotion speed was also of little importance ($\beta$ = -2.95, CrI = [-30.31, 25.03]). Running speeds ranged between 2.0 and 3.8 m/s (see also Table 1). Swing duration was not clearly affected by sex ($\beta$ = 5.41, CrI = [-13.10, 25.19]), but the locomotion speed had a moderate effect on this parameter ($\beta$ = 31.88, CrI = [1.25, 64.25]). Running cadence showed a moderate effect of sex ($\beta$ = 4.03, CrI = [-0.32, 8.47]) and speed ($\beta$ = 4.65, CrI = [-2.59, 12.06]).

In summary, the effects of sex and age on stance duration, swing duration and cadence were moderate or small and the speed at which the participants were recorded played a bigger (but more uncertainly estimated) role in walking than in running.

## Muscle synergies

The activity of 13 muscles of the lower limb (Fig 3) could be factorized into four task-related muscle synergies in all groups (Fig 4 and S2 Fig) with low to negligible effects of sex (walking:

**Table 1. Participant data and group division.** Values are presented as mean ± standard deviation.

| Group | Task | Sex | N | Age [years] | Height [cm] | Mass [kg] | Speed [m/s] |
|---|---|---|---|---|---|---|---|
| G1 | Treadmill walking | F | 35 | 25.5 ± 3.5 | 167.5 ± 6.6 | 60.0 ± 8.1 | 1.27 ± 0.12 |
| | | M | 35 | 28.3 ± 4.3 | 178.6 ± 6.1 | 75.6 ± 9.4 | 1.30 ± 0.11 |
| G2 | | F | 35 | 71.4 ± 4.9 | 165.4 ± 6.6 | 65.3 ± 9.0 | 1.11 ± 0.01 |
| | | M | 35 | 73.3 ± 4.5 | 176.7 ± 6.1 | 79.2 ± 10.4 | 1.13 ± 0.07 |
| G3 | Treadmill running | F | 60 | 28.5 ± 4.9 | 167.7 ± 6.3 | 61.0 ± 7.0 | 2.57 ± 0.28 |
| | | M | 60 | 30.5 ± 5.2 | 180.7 ± 6.0 | 76.2 ± 9.4 | 2.68 ± 0.31 |

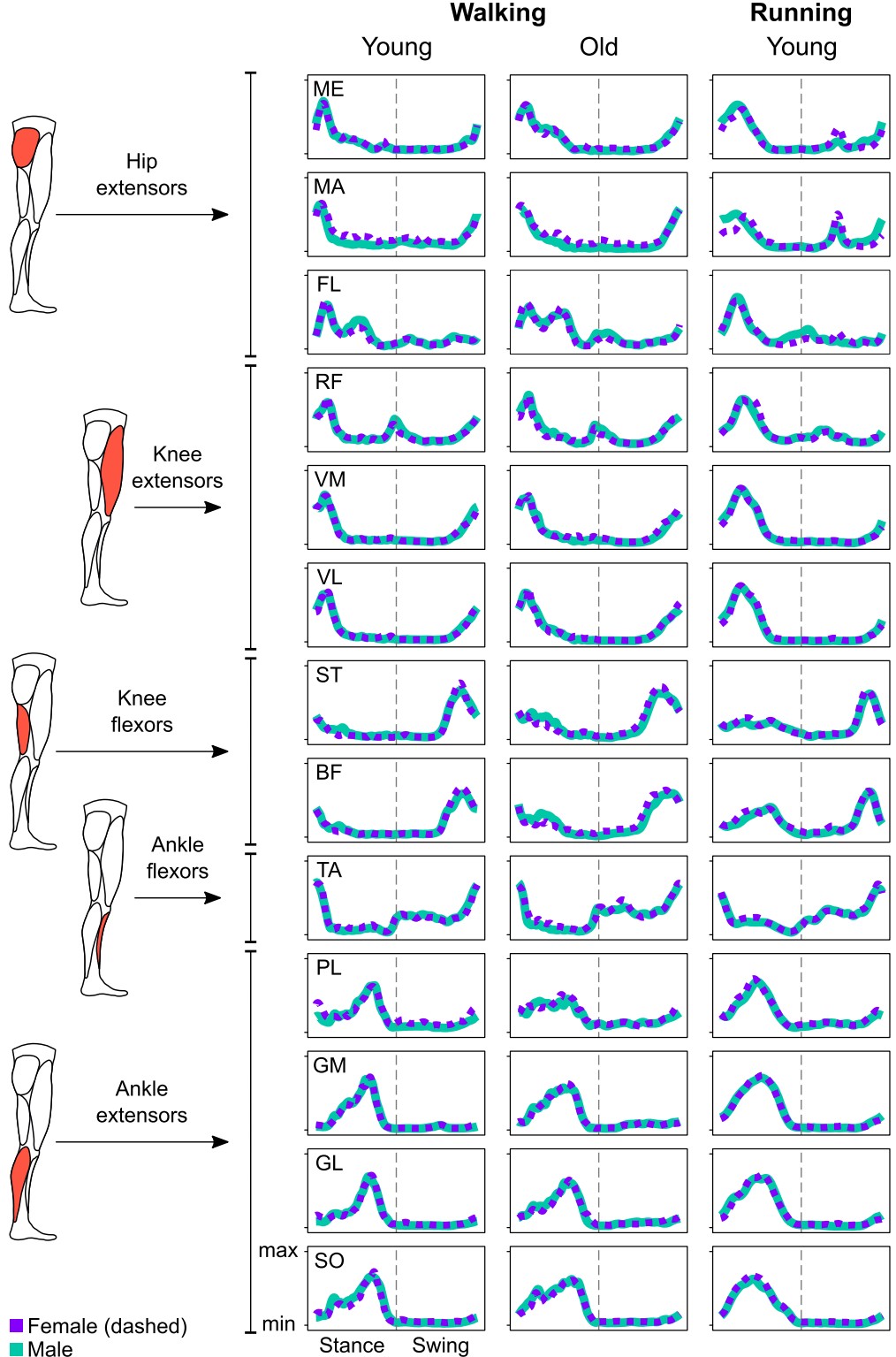

**Fig 3. Average muscle activity.** The mean activations of the recorded muscles are presented for the stance and swing phases of walking (young and old adults) and running (young adults) for female and male participants. Signals were rectified, filtered, time- and amplitude normalized as described in the methods. Muscle abbreviations: ME = gluteus medius, MA = gluteus maximus, FL = tensor fasciæ latæ, RF = rectus femoris, VM = vastus medialis, VL = vastus lateralis, ST = semitendinosus, BF = biceps femoris, TA = tibialis anterior, PL = peroneus longus, GM = gastrocnemius medialis, GL = gastrocnemius lateralis, SO = soleus.

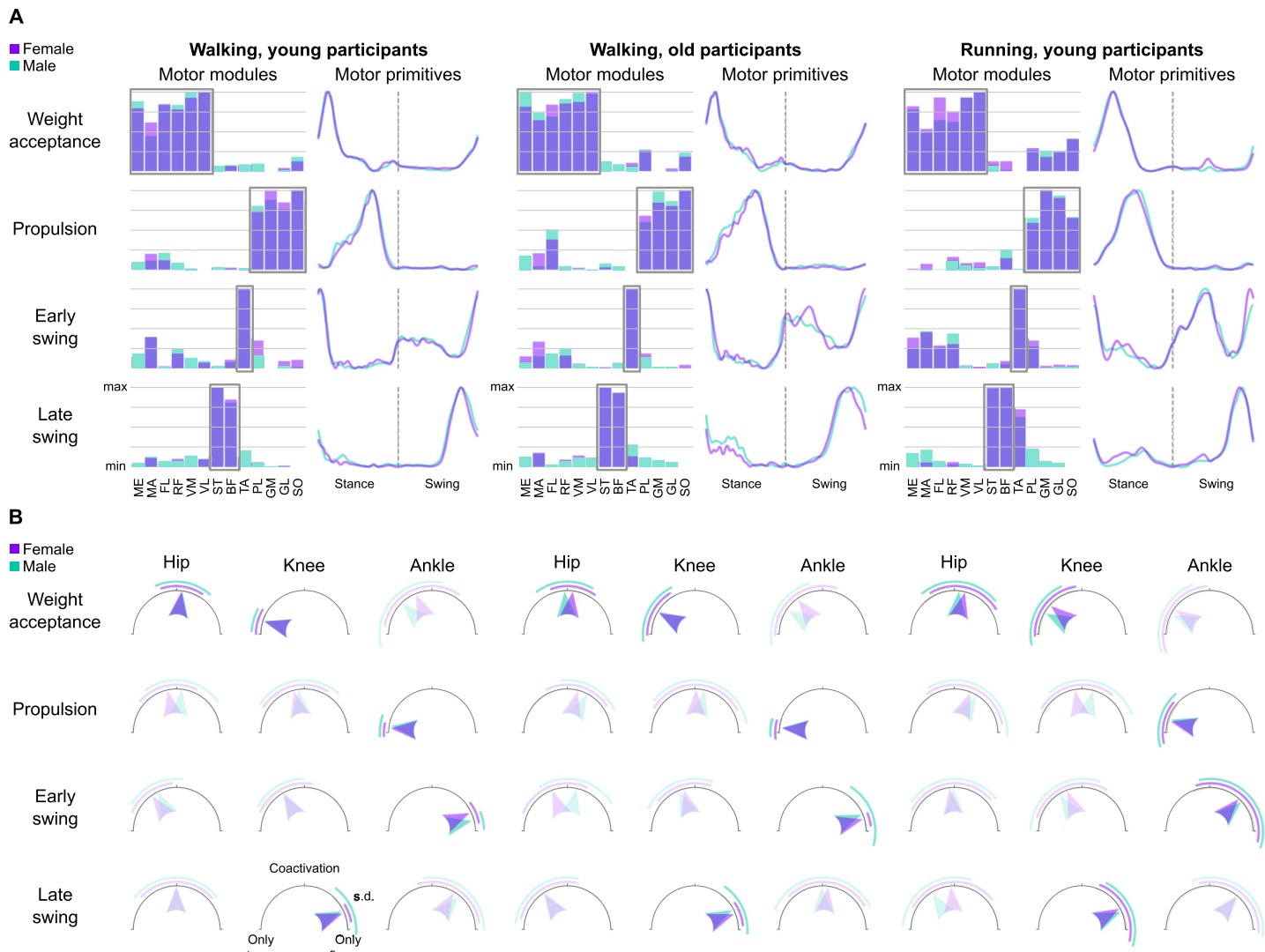

**Fig 4. Overlayed muscle synergies and coactivation index.** (A) The average muscle synergies of females and males are overlayed for comparison. The motor modules are presented on a normalized y-axis base: each muscle contribution within one synergy can range from 0 to 1 and each dot represents individual trials. Prime contributors to each synergy are highlighted by a rectangle. For the mean motor primitives (shaded standard deviation), the x-axis full scale represents the averaged gait cycle (with stance and swing normalized to the same amount of points and divided by a vertical line) and the y-axis the normalized amplitude. Muscle abbreviations: ME = gluteus medius, MA = gluteus maximus, FL = tensor fasciæ latæ, RF = rectus femoris, VM = vastus medialis, VL = vastus lateralis, ST = semitendinosus, BF = biceps femoris, TA = tibialis anterior, PL = peroneus longus, GM = gastrocnemius medialis, GL = gastrocnemius lateralis, SO = soleus. (B) Coactivation index calculated from the motor modules. Joints crossed by the most important muscles contributing to each muscle synergy are highlighted. Arrows pointing to the left show that muscle activity comes exclusively from extensors, while arrows pointing to the right mean that activity comes only from flexors. Arrows pointing upwards indicate perfect coactivation of flexors and extensors, meaning that the average module values for the flexors of that joint are the same as those for the extensors. The acronym "s. d." stands for "standard deviation".

β = 0.00, CrI = [-0.02, 0.02], $δ_t$ = 0.01; running: β = 0.01, CrI = [0.00, 0.02], $δ_t$ = 0.33) or age (β = -0.01, CrI = [-0.02, 0.00], $δ_t$ = -0.25) on the reconstruction quality ($R^2_{walking, young females}$ = 0.83 ± 0.03; $R^2_{walking, young males}$ = 0.82 ± 0.03; $R^2_{walking, old females}$ = 0.83 ± 0.03; $R^2_{walking, old males}$ = 0.82 ± 0.03; $R^2_{running, young females}$ = 0.85 ± 0.03; $R^2_{running, young males}$ = 0.85 ± 0.03). The first muscle synergy described the acceptance of the body weight, with a major involvement of the knee extensors and hip muscles. The second synergy was functionally related to the propulsion phase, mainly managed by the plantarflexors (i.e., the extensors of the ankle joint). The third

synergy represented the early stage of the swing phase, mostly orchestrated by the dorsiflexors (i.e., the ankle flexors). The fourth and chronologically last synergy was that related to the late swing phase, when mostly the knee flexors were contributing.

By looking at the motor modules (Fig 4, S2 and S3 Figs), it is possible to identify the contribution of each muscle, relative to the others, to a specific synergy. In walking, females showed a bigger contribution of the *gluteus maximus* than males in all synergies except for the late swing, where the contribution was similar. This was not the case in running, where the hip extensor was contributing similarly in both sexes in all synergies except for the late swing, were the contribution in females was lower than in males. The *gluteus medius* was more used by males during the weight acceptance of walking, while the opposite happened during the early swing in running. The *tensor fasciæ latæ* was similarly employed by both sexes during running, but in walking females showed a lower contribution of this muscle. The knee extensors behaved largely similar, but in both walking and running females used less *vastus medialis* during the weight acceptance and less *rectus femoris* during the early swing. The knee flexors were more important for the late swing phase of walking in females than in males, behaving generally similar in all the other synergies. Females used less *tibialis anterior* in the late swing synergy in walking, but used more of it during the same synergy in running. Finally, the plantarflexors were similarly managed by females and males, at least in the propulsion synergy that saw them mostly involved.

The coactivation between flexor and extensor muscles around each of the three considered joints (i.e., hip, knee and ankle, see Fig 4), revealed sparse effects of sex in both walking and running. In walking, a moderate effect of sex on coactivation was visible in the propulsion synergy, with females showing less coactivation than males ($\beta$ = -0.08, CrI = [-0.13, -0.04], $\delta_t$ = -0.37). However, the *post-hoc* analysis (S4 Fig) revealed that this difference was mostly due to the coactivation of muscles acting around the hip and the knee, two joints that are not as relevant as the ankle for the propulsion phase. Conversely, in running females showed moderately higher coactivation than males in the early swing ($\beta$ = 0.1, CrI = [0.03, 0.17], $\delta_t$ = 0.33) and late swing ($\beta$ = 0.1, CrI = [0.05, 0.15], $\delta_t$ = 0.34) synergies. While in the early swing the effect was due to the muscles acting around the joint important for that synergy (i.e., the ankle), in the late swing the differences were due mainly to the hip and ankle, two joints that were less involved than the knee in the task of stopping the leg before touchdown. The effect of ageing on coactivation had small to none interaction with sex and was of moderate effect in three out of four synergies: the propulsion ($\beta$ = 0.06, CrI = [0.02, 0.1], $\delta_t$ = 0.25), the early swing ($\beta$ = 0.07, CrI = [0.03, 0.1], $\delta_t$ = 0.36) and the late swing ($\beta$ = -0.14, CrI = [-0.18, -0.1], $\delta_t$ = -0.58).

Concerning the motor primitives, the most important effect on the width (i.e., full width at half maximum) and timing of the main activation (i.e., center of activity) during walking was the age of the participants (Figs 5 and 6). Specifically, aging had a large effect on increasing the width of all four primitives (weight acceptance: $\beta$ = 7.51, CrI = [5.29, 9.74]; propulsion: $\beta$ = 6.53, CrI = [3.36, 9.70]; early swing: $\beta$ = 12.75, CrI = [8.34, 17.13]; late swing: $\beta$ = 12.39, CrI = [9.26, 15.46]). Moreover, ageing had a moderate to large effect on shifting earlier the main activity's timing of all motor primitives (propulsion: $\beta$ = -7.24, CrI = [-9.51, -5.02]; early swing: $\beta$ = -15.48, CrI = [-23.78, -7.01]; late swing: $\beta$ = -17.14, CrI = [-27.42, -6.6]) except the weight acceptance ($\beta$ = 5.32, CrI = [-3.44, 14.3]). However, the sex of the participants did not result in such large changes. Females showed a moderately narrower (i.e., shorter) propulsion primitive ($\beta$ = -3.06, CrI = [-6.98, 0.81]). A *post-hoc* investigation of the yet weak interaction of sex and age showed that the narrowing in females mostly happened in young participants and not in older ones (S5 Fig). Moreover, sex had a moderate effect on the timing of the early swing primitive, which in females was located earlier in time than in males ($\beta$ = -10.61, CrI = [-21.32, 0.06]). In running, sex had a small effect on the width of the weight acceptance primitive,

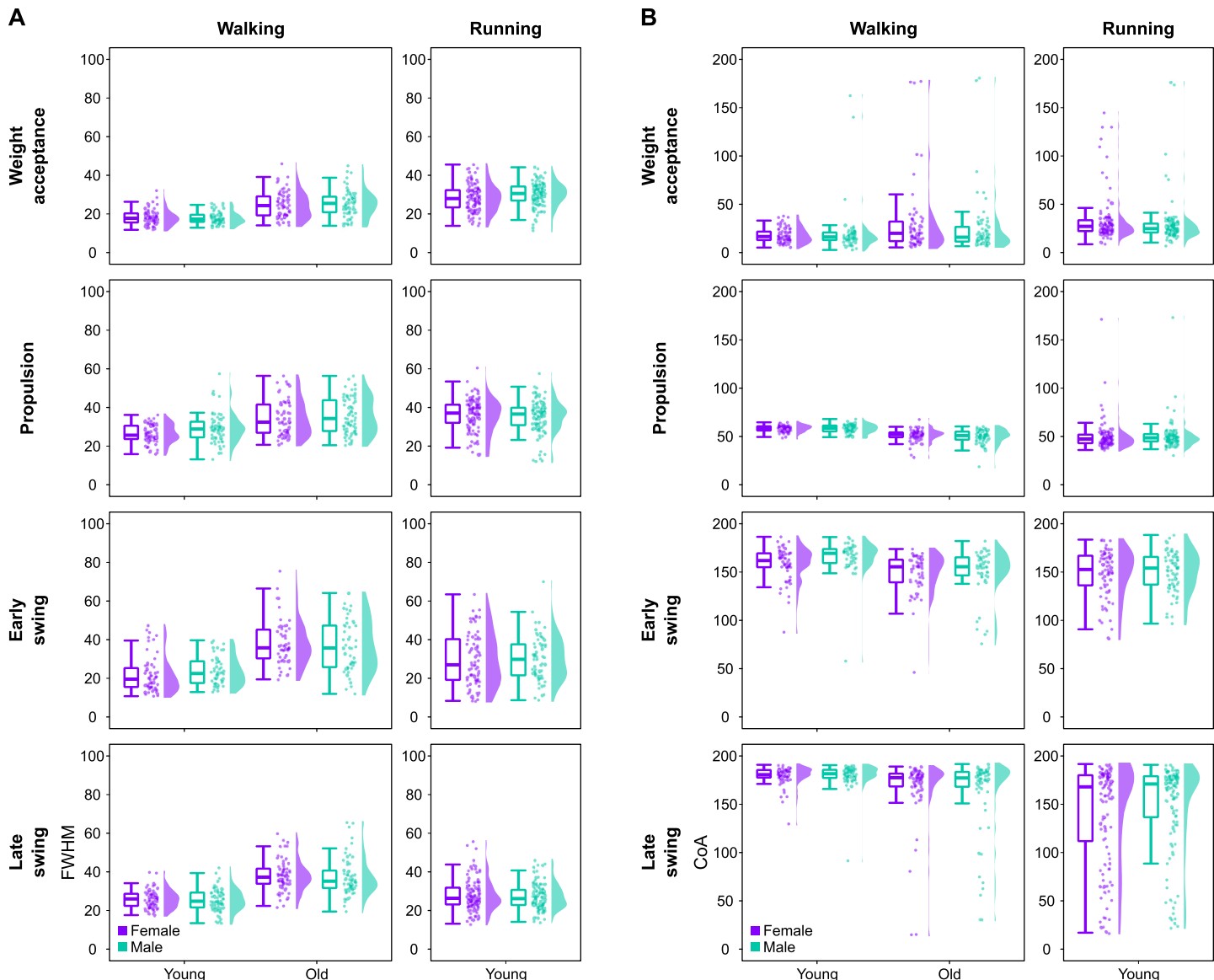

**Fig 5. Full width at half maximum and center of activity of motor primitives.** (A) Boxplots describing the full width at half maximum (FWHM) of the extracted motor primitives. Raw data points (dots represent the mean value of 30 gait cycles from every participant) and their density estimates are presented to the right side of each boxplot. (B) The same as in panel (A), but for the center of activity (CoA).

which was reduced in females (β = -1.49, CrI = [-3.66, 0.69]). All the other effects were smaller and the same considerations done above for the effect of the different locomotion speeds are valid also for the full width at half maximum and center of activity of motor primitives.

In summary, the analysis of motor modules revealed largely similar relative muscle contributions in females and males, but with some sparse differences involving mostly the hip extensors, knee extensors and foot dorsiflexors. Likewise, we found sparse sex-related differences on the coactivation of synergistic muscles, while motor primitives resulted wider in males in the propulsion synergy for walking (but only in young and not in older adults) and in the weight acceptance synergy for running.

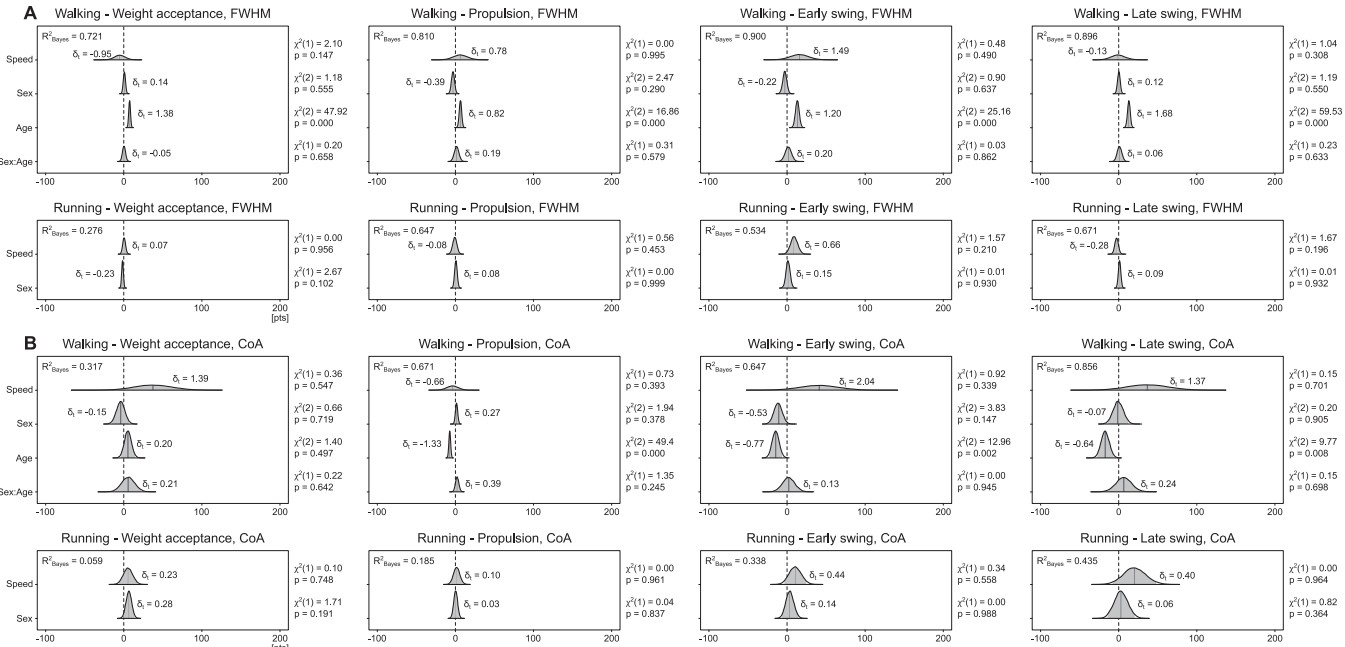

**Fig 6. Posterior 95% credible intervals for the estimated differences in full width at half maximum and center of activity.** (A) The 95% credible intervals and their probability distributions (shaded areas) describe the effects and interaction of sex and age (with locomotion speed as a covariate) on the full width at half maximum (FWHM). Positive values indicate wider primitives in females than in males and in older than in younger adults. Effect size in the style of Hedges (i.e., considering all the variance sources in the model) are shown on the graphs and called $\delta_t$. A Bayesian equivalent of $R^2$ (here $R^2_{Bayes}$) describes how well the model fits the data. Next to each estimation, the $\chi^2$ and p-values resulting from a frequentist linear mixed model analysis are reported as additional information. (B) The same as in panel (A), but for the center of activity (CoA).

## Muscle synergy dynamics

The fractal analysis of motor primitives did not reveal any large effect of either age or sex on the local (i.e., Higuchi's fractal dimension) and global (i.e., Hurst exponent) complexity (Figs 7 and 8). In walking, sex had a small effect on both the local and global complexity, reducing them in females (local: β = -0.01, CrI = [-0.04, 0.01]; global: β = -0.01, CrI = [-0.05, 0.02]). However, we could observe a moderate interaction between sex and age (local: β = -0.04, CrI = [-0.08, 0.00]; global: β = -0.04, CrI = [-0.08, 0.01]), the *post-hoc* analysis of which revealed that older females tended to reduce more than the younger counterparts both the local and global complexity as compared to males (S6 Fig).

In summary, young adults did not show sex-related differences in the local and global complexity of motor primitives, but this changed in older adults indicating that aging might influence the way females and males modulate the dynamics of motor primitives.

## Discussion

The investigation of sex as a biological variable is of crucial importance for the equitable assessment of human performance, pathology, aging effects, and so forth. Here, we investigated the effects of sex—and how these may be influenced by aging—on the modular organization of muscle activity during locomotion. We hypothesized and subsequently found that, in general, there is high similarity between sexes in gait timing and synergistic muscle activity. The sparse but potentially important sex-specific modulations of motor control found in this study might

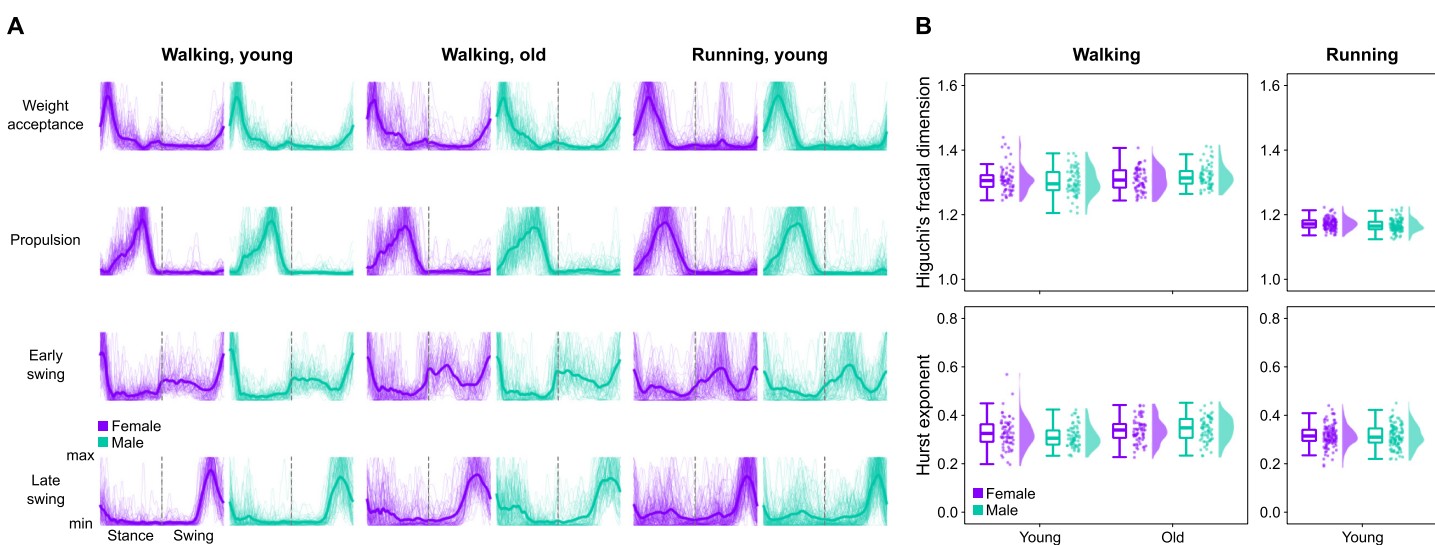

**Fig 7. Motor primitives and their fractal properties.** (A) Overall average motor primitives (thick lines) and each participant's average motor primitive (thin lines). (B) Boxplots describing the Higuchi's fractal dimension and the Hurst exponent of the extracted motor primitives. Raw data points (dots represent the mean value from every participant) and their density estimates are presented to the right side of each boxplot.

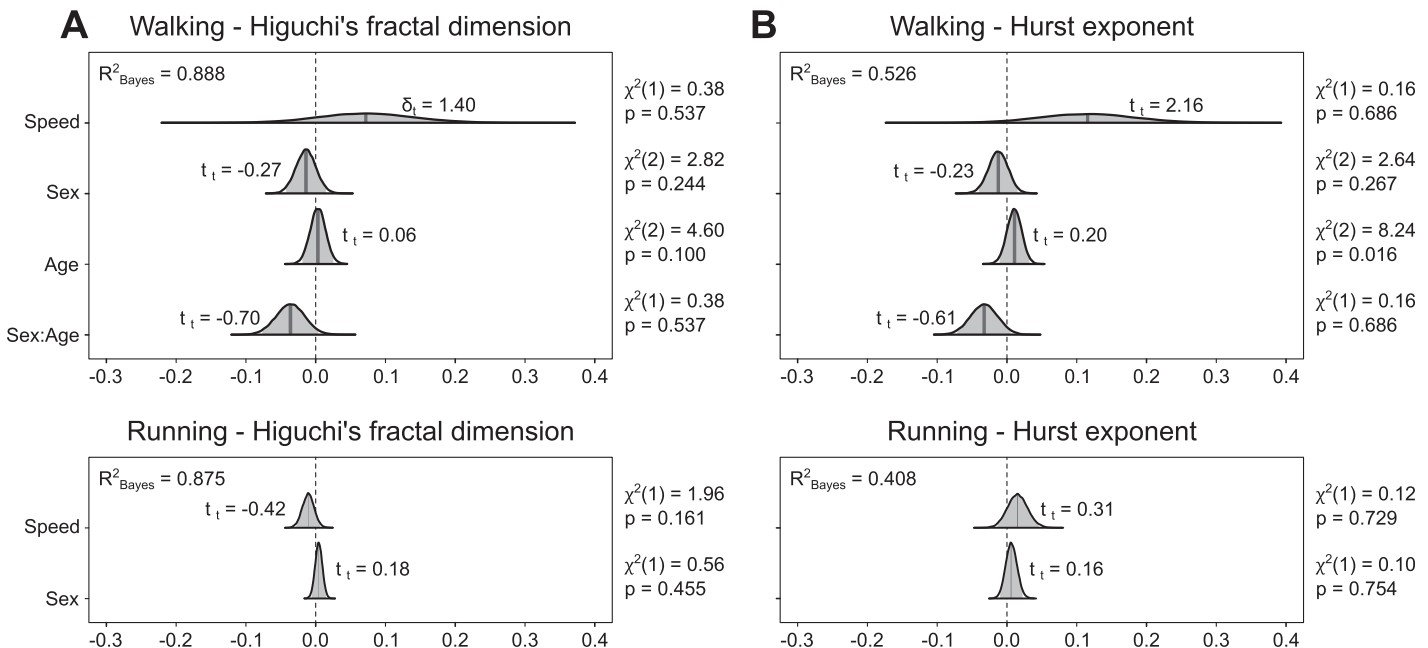

**Fig 8. Posterior 95% credible intervals for the estimated differences in Higuchi's fractal dimension and Hurst exponent.** (A) The 95% credible intervals and their probability distributions (shaded areas) describe the effects and interaction of sex and age (with locomotion speed as a covariate) on the Higuchi's fractal dimension. Positive values indicate higher fractal dimension in females than in males and in older than in younger adults. Effect size in the style of Hedges (i.e., considering all the variance sources in the model) are shown on the graphs and called $\delta_t$. A Bayesian equivalent of $R^2$ (here $R^2_{Bayes}$) describes how well the model fits the data. Next to each estimation, the $\chi^2$ and p-values resulting from a frequentist linear mixed model analysis are reported as additional information. (B) The same as in panel (A), but for the Hurst exponent.

be leveraged in the future to deepen the neurophysiological mechanisms that regulate sex-related diversity in the response to exercise.

When walking on a treadmill at a speed close to the average preferred one, females showed shorter swing times than males, regardless of age. Older adults showed shorter swing and longer stance times than younger participants, thus the overall cadence was not affected. In running, where we only analyzed younger adults, sex did not play a big role on the stance and swing times, while cadence was moderately higher in females than in males. This first part of the results shows that, in the considered participants, sex-related differences were limited to the swing times in walking and cadence in running and they did not vary with aging, largely in agreement with previous literature [11, 13, 22, 40].

The four synergies that described the phases of weight acceptance, propulsion, early swing and late swing were found independently of sex, age group or locomotion types, in accordance with most of the available body of existing literature that, however, does not include any study that specifically investigated sex as a biological variable during locomotion [31, 33, 37, 41, 42]. Some previous works that analyzed non-factorized EMG activities from the lower limbs found that females and males shared similar muscle activations during both walking [12–17] and running [18, 19]. However, subtler sex-related differences were found in other works. For instance, females have been known to show higher hip extensor, knee extensor, foot dorsi-flexor or foot plantarflexor activity than males during walking [20–22] or running [23–25]. Although motor modules, due to data normalization, describe the relative contribution of one muscle with respect to the others in a certain synergy and not the absolute amplitude of the signal, our analysis could partially confirm those previous findings. The identified sex-specific contributions of the hip, knee and ankle joint muscles to the relevant synergies were sparse. However, if further investigated, they could potentially be useful for the design of locomotion-based experimental protocols or training interventions that account for the biological sex of the participants. On a related note, it has been previously reported that females might show more preactivation than males in muscles such as the knee flexor *biceps femoris* in walking [19] and the foot evertor and plantarflexor *peroneus longus* in running [26]. The two muscles are respectively contributing to the propulsion synergy for walking and late swing synergy for running. Preactivation is a time-related strategy, defined as an activation happening before the functional use of the muscle. However, our analysis of the width and main peak timing revealed that the propulsion primitive for walking and the weight acceptance primitive for running were narrower, but not time-shifted, in females than in males. Thus, our results do not support a preactivation of the knee flexors (mainly important for the late swing synergy) in walking and of foot plantarflexors (mainly important for the propulsion synergy) in running, at least at a muscle-group level. A narrowing without modification of the main peak timing is yet an interesting outcome that implies shorter—and not time-shifted—basic activation patterns in females than in males in two synergies that occur during stance. This finding is additionally interesting given the high similarity of the stance duration between the two sexes, implying that the narrowing happened in absolute terms (i.e., given the same stance duration, females spent less time than males in the propulsion phase of walking and weight acceptance phase of running). An explanation of this phenomenon cannot be directly supported by our results, but it is possible to speculate on its potential neurophysiological importance based on present and past outcomes.

The temporal widening of motor primitives, but also of muscle-specific activation patterns, has been previously interpreted as a compensatory mechanism adopted by the central nervous system to cope with the challenges imposed by: a) internal perturbations deriving from pathology [43–45] or sensory feedback impairment [46], b) external mechanical perturbations [31, 47–49], c) high locomotion speeds [50], d) development stage [38, 45] or e) aging [30, 31], the

latter being confirmed by the results of the present study. Here, we recruited healthy, young adults of both sexes and had them walk or run at similar submaximal speeds. It is thus unlikely that the sex-specific width of the stance phase primitives was linked to any of the aforementioned factors. Our results show that muscles were being activated for a shorter amount of time in females than in males during the stance phase of locomotion, when muscle activities are at their highest (and so are energy requirements). Yet, it is known from previous studies that, in elite athletes running at similar submaximal speeds, males are more economical than females in terms of absolute oxygen uptake and no sex specificity exists when the oxygen uptake is normalized by the body mass [51]. Thus, the shorter activation does not translate in lower energy consumption, likely due to the myriad of other variables that come into play. Moreover, despite previous sporadic findings of higher muscle coactivation in females than in males during walking [27, 28], here we showed that females had lower coactivation at the ankle joint during the propulsion phase of walking, while during running the weight acceptance showed similar coactivation levels. However, females are known to possess superior resistance to fatigue after prolonged whole-body submaximal exercise [4, 52–55], thus the narrowing of motor primitives (i.e., reduced utilization over time) associated with the demanding stance phase of locomotion might constitute a neuromotor strategy to limit fatigability during prolonged locomotion efforts.

With our analysis, we could also show that sex specificities were generally maintained through aging, but not in all parameters. For instance, the shorter stance phase in males than in females was more evident in participants of older age than in the younger ones. Conversely, the narrower propulsion primitive for walking in females was more evident in younger than in older participants. Last but not least, both the local and global complexity of motor primitives were lower in older females than in older males, but this would not hold for the younger participants. These outcomes indicate that sex-specific strategies are not necessarily retained through aging, pointing to the possibility that the course of motor optimization over lifespan is sex-dependent and needs to be considered when designing therapy or exercise protocols for quality-of-life improvement.

The presented results must be interpreted with an eye to the limitations of the design. Participants were not tested in the same experimental environment and three distinct laboratories and treadmills—but the same EMG system—were used to gather the data. Moreover, due to different experimental setups, some participants walked or ran at their preferred speed, while others were asked to complete the locomotor task at speeds that were predetermined by the researchers. While the difference between preferred and predetermined speeds were small, we cannot exclude an effect on our results. Last but not least, the groups were formed to be as homogeneous as possible. Yet, small to large effects of sex were observed in both age and body mass index, to which the presented results were not normalized.

With those limitations in mind, the main results can be summarized as follows:

- Neither sex or age had large effects on stance, swing or cadence during walking or running

- Relative muscle contributions (i.e., motor modules and coactivations) were largely unaffected by sex and age, with minor and sparse differences

- Muscle activation motifs (i.e., motor primitives) were wider in younger males in the propulsion synergy (walking) and in the weight acceptance synergy (running); in older adults, this difference disappeared

- The complexity of motor primitives for walking was affected by sex in older (i.e., lower in females) but not in younger adults.

In conclusion, the albeit subtle sex-specific modulations of motor control found in this study justify on several levels the need to continue designing research protocols based on mixed cohorts [5]. Physiological discrepancies between males and females are known to affect the response to acute and long-term exercise and it is likely that interventions designed based on data coming from male participants might not fit the physiology of both sexes [4]. This is also true for the diagnosis of a great amount of medical conditions: the descriptions of symptoms are historically largely based on data coming from male participants, leading to increased likelihood of misdiagnosis in females [1]. As Shansky and Murphy recently advocated, representing both sexes instead of just one is a more rigorous approach that can improve reproducibility and translatability, potentially allowing for the discovery of "sex-dependent mechanisms underlying a common outcome" [1].

## Materials and methods

This study was reviewed and approved by the Ethics Committees of the Humboldt-Universität zu Berlin (approval code HU-KSBF-EK_2018_0013), Kassel University (E05201602) and Heidelberg University (AZ Schw 2018 1/2). All the participants gave written informed consent for the experimental procedure, in accordance with the Declaration of Helsinki.

### Experimental protocols

We recruited 215 healthy volunteers, 70 of which were of older age (65 to 84 years). The young adults (20 to 43 years) were asked to walk and/or run on a treadmill, while the older adults were only asked to perform treadmill walking. Three experimental conditions were obtained as follows: walking, 70 young adults (henceforth G1, 35 females); walking, 70 older adults (henceforth G2, 35 females); running, 120 young adults (henceforth G3, 60 females; some participants in G3 were also included in G1, the specifics are given in the supplementary information at Zenodo [56]. Details about the participants and locomotion speeds are given in Table 1: females were younger (10.0% in G1, Cohen's $d$ = -0.72; 2.6% in G2, $d$ = -0.40; 6.3% in G3, $d$ = -0.37) and had lower body mass index (9.6% in G1, $d$ = -0.86; 5.9% in G2, $d$ = -0.53; 6.5% in G3, $d$ = -0.65) than males. The participants completed a self-selected warm-up walking (G1, G2) or running (G1, G3) on a treadmill, typically lasting between 3 and 5 min [57]. The participants were asked to walk or run at either their preferred speed, found through the method of limits [58], or at a fixed speed chosen to be close to the average preferred speed, depending on the group [57, 59] (details are given in the supplementary information at Zenodo [56]). The treadmills used were of three different types, depending on the location of the recordings: a) mercury, H-p-cosmos Sports & Medical GmbH, Nussdorf, Germany equipped with a pressure plate recording the plantar pressure distribution at 120 Hz (FDM- THM-S, zebris Medical GmbH, Isny im Allgäu, Germany) for the experiments conducted at the Humboldt-Universität zu Berlin; b) Laufergotest, Erich Jäger, Würzburg, Germany for the experiments at Kassel University; c) BalanceTutor™, MediTouch LTD, Netanya, Israel, for the experiments at Heidelberg University. Data collection for the participants of group G1 was conducted on treadmills a) (33 participants, 16 females), b) (18 participants, 7 females) and c) (19 participants, 12 females); for group G2 on treadmill c); for group G3 on treadmills a) (105 participants, 53 females) and b) (15 participants, 7 females). Part of the presented data sets have been previously published: G1 partially in [31, 41, 47, 59]; G2 partially in [31]; G3 partially in [31, 41, 47, 59, 60].

### EMG recordings

The muscle activity of the following 13 muscles of the right side was recorded in all groups: *gluteus medius*, *gluteus maximus*, *tensor fasciæ latæ*, *rectus femoris*, *vastus medialis*, *vastus lateralis*,

*semitendinosus*, *biceps femoris* (long head), *tibialis anterior*, *peroneus longus*, *gastrocnemius medialis*, *gastrocnemius lateralis* and *soleus*. The electrodes were positioned as previously reported and following SENIAM (surface EMG for a non-invasive assessment of muscles) recommendations [59, 61]. After around 60 s habituation [47], we recorded two trials of at least 35 gait cycles for each participant with an acquisition frequency of 1 kHz by means of a 16-channel wireless bipolar EMG system (Wave Plus wireless EMG with PicoEMG sensors, Cometa srl, Bareggio, Italy). For the EMG recordings, we used foam-hydrogel electrodes with snap connector (H124SG, Medtronic plc, Dublin, Ireland). The first 30 gait cycles of the recorded trial were considered for subsequent analysis [62]. All the recordings can be downloaded from the supplementary data set, accessible at Zenodo [56].

## Gait cycle parameters

The gait cycle segmentation (foot touchdown and lift-off timing) was obtained by the elaboration of the data acquired by either the pressure plate integrated in one of the treadmills (3 participants in G1 and 11 in G3), a foot-mounted 3D accelerometer (PicoEMG, Cometa srl, Bareggio, Italy) recording at 142.85 Hz (49 participants in G1, all participants in G2 and 94 participants of G3) or 3D kinematics of the foot recorded via a six-camera motion capture system (Oqus 3+, Qualisys AB, Gothenburg, Sweden) acquiring at 300 Hz (18 participants in G1 and 15 in G3). For the detection of both touchdown and lift-off of the foot, we used externally-validated algorithms based on foot acceleration data. Specifically, for touchdown we adopted the modified foot contact algorithm [63], which uses a characteristic peak in the vertical acceleration to determine touchdown. For lift-off, we searched for a characteristic peak in the longitudinal (i.e., anteroposterior) acceleration of the foot between two consecutive touchdowns. Both algorithms were validated against data recorded from 30 young adults (15 females, height 173 ± 10 cm, body mass 68 ± 12 kg, age 28 ± 5 years) during walking and running (at 1.4 m/s and 2.8 m/s, respectively) over a 900 x 600 mm force plate (1 kHz, AMTI BP600, Advanced Mechanical Technology, Inc., Watertown, MA, USA), with ten steps recorded per participants, for a total of 300 steps for walking and 300 for running. For the 14 participants in which plantar pressure distribution was recorded, we elaborated this data to obtain the gait cycle segmentation [57]. Gait temporal parameters calculated from either the pressure plate, foot accelerations or motion capture were the cadence (i.e., number of steps per minute), stance and swing times (in milliseconds).

## Muscle synergies extraction

Muscle synergies data were extracted from the recorded EMG activity through a custom script (R v4.0.4, R Core Team, 2021, R Foundation for Statistical Computing, Vienna, Austria) based on the R package "musclesyneRgies" [64] v0.7.1-alpha and using the classical Gaussian NMF algorithm [65], the application of which was extensively described in the past [46, 47, 66, 67]. Briefly, we high-pass filtered, full-wave rectified and lastly low-pass filtered the raw EMG signals using a $4^{th}$ order IIR Butterworth zero-phase filter with cut-off frequencies 50 Hz (high-pass) and 20 Hz (low-pass). After amplitude normalization of each muscle to the maximum of each trial, gait cycles were time-normalized to 200 points each (100 points for the stance, 100 points for the swing phase). Synergies were then extracted through NMF and the quality of reconstruction assessed with the coefficient of determination $R^2$. Similar time-dependent motor primitives [38, 59] and time-invariant motor modules [59, 68] were then functionally classified through a clustering approach based on NMF [41]. The source code for the pre-processing of raw EMG data and for the extraction and classification of muscle synergies is

available at https://github.com/alesantuz/musclesyneRgies [69]. Version 0.7.1-alpha, used in this paper, is archived at Zenodo [56].

## Coactivation index

To obtain the ratio of flexor and extensor muscle contribution to each joint in a specific synergy, we calculated the coactivation index (CaI) from the motor modules as follows. For every synergy and trial, we calculated the mean contributions of the flexors and those of the extensors. For the hip, we considered the FL and RF as flexors and the ME and MA as extensors. For the knee, the flexors were the ST and BF and the extensors the RF, VM and VL. For the ankle, the only flexor (i.e., foot dorsiflexor) was the TA and the extensors (i.e., foot plantarflexors) the PL, GM, GL and SO. For each joint, the mean of the flexor contributions $\overline{flex}$ and the mean of the extensor contributions $\overline{ext}$ were forced to sum to 1: $CaI = \overline{flex}/(\overline{flex} + \overline{ext})$. Following this definition: a) CaI = 0 when only extensors are contributing to the considered joint; b) CaI = 1 when only flexors are giving their contribution; c) CaI = 0.5 if flexors and extensors are equally contributing (i.e., full coactivation of flexors and extensors).

## Linear and nonlinear metrics for the analysis of motor primitives

As linear metrics for the comparison of motor primitives we used the full width at half maximum and the center of activity. The full width at half maximum is used to describe the duration (i.e., the width) of motor primitives relative to the gait cycle and it was calculated cycle-by-cycle after subtracting the cycle's minimum as the number of points exceeding each cycle's half maximum, and then averaged [43]. The center of activity, an indication of when the main activation is happening in time, was also calculated cycle-by-cycle as the angle of the vector in polar coordinates that points to the center of mass of the circular distribution defined between 0 (touchdown) and 360˚ (next touchdown) and then averaged [45]. The advantage of using the center of activity instead of the global maximum is that the former can better locate the timing of main activity when primitives are "rough" and several local maxima approaching the value of the global maximum are present in the signal [45].

To assess the local and global complexity of motor primitives, we calculated the Higuchi's fractal dimension and Hurst exponent, respectively [70, 71]. The local complexity can be seen as a measure of "roughness" (or noise content) in the signal within each gait cycle (thus the term "local"), while the global complexity is a measure of how "accurate" each cycle's activations are, when compared to the others of the same trial (thus the term "global"). The numerical procedures for obtaining both metrics were recently reported in detail [72]. The Higuchi's fractal dimension was calculated with a maximum window size of 10 points, while the Hurst exponent was obtained by imposing a minimum window length of 200 points, equal to the length of a time-normalized gait cycle. Values of the Higuchi's fractal dimension range from 1 to 2, with increasing values correlating to increasingly complex (or rough) data and 1.5 indicating random Gaussian noise [70, 73, 74]. The Hurst exponent can vary between 0 and 1. For 0.5 < Hurst exponent < 1, in the long-term high values in the time series (the motor primitive in our case) will be probably followed by other high values and a positive or negative trend is visible [75, 76]. For 0 < Hurst exponent < 0.5, in the long term high values in the series will be probably followed by low values, with a frequent switch between high and low values [75, 76] and values moving from 0.5 towards 0 indicating increasingly stereotyped gait cycle's activity (e.g., a sinusoidal wave has Hurst exponent = 0 and each cycle is identical to the others). A completely random series will have Hurst exponent = 0.5 [76, 77].

## Statistics

To investigate the effects of sex on the modular muscle activation, we followed a Bayesian multi-level modeling approach implemented in the R package *brms* 2.15.0 [78, 79]. We adopted the Bayesian framework in an effort to move away from the concept of statistical significance and its dichotomous nature, including in our analysis prior data and available information for computing credible intervals around the observed effects [80, 81]. Credible intervals are "the Bayesian analogue of a classical confidence interval" [82], with the advantage that it is possible to make probability statements on them: a 95% credible intervals can be defined as that interval around the point estimate having a 95% probability of encompassing the population value, given the data and the prior assumptions [82]. For each dependent variable of interest, we built a mixed effects model containing both "fixed" and "random" effects. The constant effects analyzed were: sex, age and their interaction in G1 and G2 (Eq 1); sex in G3 (Eq 2). Locomotion speed was modeled as a covariate in all groups, to account for the different speeds between groups and/or participants. As random effects, we considered by-participant random slopes for the effect of the independent variables. The two models can be respectively summarized with the following equations:

$$y \sim speed + sex * age + (1|participant) \tag{1}$$

$$y \sim speed + sex + (1|participant) \tag{2}$$

where $y$ is the analyzed dependent variable. Each model was run with five independent Markov chains of 5000 iterations, with the first 2000 warm-up iterations used only for calibration and then discarded, thus resulting in 15000 post-warm-up samples. Sex was contrast-coded symmetrically (female = 0.5, male = -0.5). Convergence of the chains and sufficient sampling of posterior distributions were confirmed by ensuring a potential scale reduction factor $\hat{R} < 1.01$ and an effective sample size of at least 20% of the number of iterations [83]. Posterior distributions were summarized by the following estimators: mean, standard error and the lower and upper bounds of the 95% credible intervals. We used different priors depending on the investigated parameters and based on previously published data [31, 47, 50, 59, 60], without distinction of sex or age and doubling the standard deviation for normal distributions and using scale $\theta = 0.5$, shape $k$ = mean $/ \theta$ for gamma distributions. Specifically, we used a gamma prior for the factorization rank with parameters shape and scale: rank $\sim \Gamma(9, 0.5)$. For all the other parameters, we used normal priors with the following means and standard deviations: stance duration walking $\sim$ N(700 ms, 100 ms); stance duration running $\sim$ N(300 ms, 80 ms); swing duration walking $\sim$ N(400 ms, 70 ms); swing duration running $\sim$ N(450 ms, 100 ms); cadence walking $\sim$ N(110 steps/min, 15 steps/min); cadence running $\sim$ N(165 steps/min, 25 steps/min); percentage of combined synergies $\sim$ N(20%, 10%); $R^2 \sim$ N(30, 15); FWHM $\sim$ N(30 points, 15 points); CoA $\sim$ N(100 points, 35 points); Higuchi's fractal dimension $\sim$ N(0.30, 0.15); Hurst exponent $\sim$ N(1.3, 0.2); CaI $\sim$ N(50%, 15%). In the results section we reported the slope estimated by the statistical model as β and the estimated 95% credible interval as CrI. To report the effect size, we calculated the Bayesian equivalent of the Cohen's $d$, called $\delta_t$ [82]. Moreover, we expressed a Bayesian version of the $R^2$, here called $R^2_{Bayes}$ and calculated as the variance of the predicted values divided by the variance of predicted values plus the expected variance of the errors, to quantify the model's fit to the original data, [84]. Values of $\delta_t$ and $R^2_{Bayes}$ are to be interpreted similarly to the Cohen's $d$ and the classical $R^2$, respectively.

## Supporting information

**S1 Fig.** *Post-hoc* **95% credible intervals for the estimated differences in stance duration between females (F) and males (M).** The 95% credible intervals and their probability

distributions (shaded areas) describe the *post-hoc* analysis of the interaction between sex and age on the stance duration. Negative values indicate shorter stance times in females than in males.
(EPS)

**S2 Fig. Muscle synergies.** The motor modules are presented on a normalized y-axis base: each muscle contribution within one synergy can range from 0 to 1 and each dot represents individual trials. For the mean motor primitives (shaded standard deviation), the x-axis full scale represents the averaged gait cycle (with stance and swing normalized to the same amount of points and divided by a vertical line) and the y-axis the normalized amplitude. Muscle abbreviations: ME = gluteus medius, MA = gluteus maximus, FL = tensor fasciæ latæ, RF = rectus femoris, VM = vastus medialis, VL = vastus lateralis, ST = semitendinosus, BF = biceps femoris, TA = tibialis anterior, PL = peroneus longus, GM = gastrocnemius medialis, GL = gastrocnemius lateralis, SO = soleus.
(EPS)

**S3 Fig. *Post-hoc* 95% credible intervals for the estimated differences in motor modules between females (F) and males (M).** The 95% credible intervals and their probability distributions (shaded areas) describe the *post-hoc* analysis of the interaction between sex and muscle on the motor modules. Positive values indicate higher muscle contribution in females than in males. Muscle abbreviations: ME = *gluteus medius*, MA = *gluteus maximus*, FL = *tensor fasciæ latæ*, RF = *rectus femoris*, VM = *vastus medialis*, VL = *vastus lateralis*, ST = *semitendinosus*, BF = *biceps femoris*, TA = *tibialis anterior*, PL = *peroneus longus*, GM = *gastrocnemius medialis*, GL = *gastrocnemius lateralis*, SO = *soleus*.
(EPS)

**S4 Fig. *Post-hoc* 95% credible intervals for the estimated differences in coactivation between females (F) and males (M).** The 95% credible intervals and their probability distributions (shaded areas) describe the *post-hoc* analysis of the interaction between sex and joint on the coactivation index. Joints crossed by the most important muscles contributing to each muscle synergy are highlighted. Negative values indicate lower coactivation index in females than in males (please note that coactivation index values close to zero mean that muscle activity comes exclusively from extensors; values close to one mean that activity comes only from flexors; values of 0.5 indicate perfect coactivation of flexors and extensors, meaning that the average module values of flexors for that joint are the same as the extensors').
(EPS)

**S5 Fig. *Post-hoc* 95% credible intervals for the estimated differences in full width at half maximum between females (F) and males (M).** The 95% credible intervals and their probability distributions (shaded areas) describe the *post-hoc* analysis of the interaction between sex and age on the full width at half maximum (FWHM). Negative values indicate narrower primitives in females than in males.
(EPS)

**S6 Fig. *Post-hoc* 95% credible intervals for the estimated differences in Higuchi's fractal dimension and Hurst exponent between females (F) and males (M).** (A) The 95% credible intervals and their probability distributions (shaded areas) describe the *post-hoc* analysis of the interaction between sex and age on the Higuchi's fractal dimension. Positive values indicate higher fractal dimension in females than in males. (B) The same as in panel (A), but for the Hurst exponent.
(EPS)

## Acknowledgments

The authors are grateful to all the participants and disclose any professional relationship with companies or manufacturers who might benefit from the results of the present study.

## Author Contributions

**Conceptualization:** Alessandro Santuz, Juri Taborri, Adamantios Arampatzis.

**Data curation:** Alessandro Santuz.

**Formal analysis:** Alessandro Santuz.

**Investigation:** Alessandro Santuz, Leon Brüll, Victor Munoz-Martel.

**Methodology:** Alessandro Santuz.

**Project administration:** Adamantios Arampatzis.

**Resources:** Adamantios Arampatzis.

**Software:** Alessandro Santuz.

**Supervision:** Adamantios Arampatzis.

**Validation:** Alessandro Santuz.

**Visualization:** Alessandro Santuz.

**Writing – original draft:** Alessandro Santuz, Lars Janshen, Leon Brüll, Victor Munoz-Martel, Adamantios Arampatzis.

**Writing – review & editing:** Alessandro Santuz, Lars Janshen, Leon Brüll, Victor Munoz-Martel, Juri Taborri, Stefano Rossi, Adamantios Arampatzis.

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
