## [Decision Letter · Decision Letter 0]

13 Dec 2021

PONE-D-21-25840Sex-specific tuning of modular muscle activation patterns for locomotion in young and older adultsPLOS ONE

Dear Dr. Santuz,

Thank you for submitting your manuscript to PLOS ONE. After careful consideration, we feel that it has merit but does not fully meet PLOS ONE’s publication criteria as it currently stands. Therefore, we invite you to submit a revised version of the manuscript that addresses the points raised during the review process. The topic of your paper is interesting, however, there are a number of questions that remain open. Due to the unusual statistics, a third reviewer was invited to specifically focus on the statistical analysis. Please respond to these comments very carefully.

We look forward to receiving your revised manuscript.

Kind regards,

Peter Andreas Federolf

Academic Editor

PLOS ONE

Journal Requirements:

[The authors are grateful to all the participants and acknowledge support by the German Research Foundation (DFG) and the Open Access Publication Fund of the Humboldt-Universität zu Berlin. The authors disclose any professional relationship with companies or manufacturers who might benefit from the results of the present study.]

 [The authors received no specific funding for this work.]

Reviewers' comments:

Reviewer's Responses to Questions

**Comments to the Author**

1. Is the manuscript technically sound, and do the data support the conclusions?

Reviewer #1: Yes

Reviewer #2: Yes

Reviewer #3: Partly

2. Has the statistical analysis been performed appropriately and rigorously? 

Reviewer #1: Yes

Reviewer #2: I Don't Know

Reviewer #3: No

3. Have the authors made all data underlying the findings in their manuscript fully available?

Reviewer #1: Yes

Reviewer #2: Yes

Reviewer #3: Yes

4. Is the manuscript presented in an intelligible fashion and written in standard English?

Reviewer #1: Yes

Reviewer #2: Yes

Reviewer #3: Yes

5. Review Comments to the Author

Reviewer #1: The present manuscript uses the non-negative matrix factorization technique to assess potential differences in synergistic muscle activation patterns in walking and running between men and women as well as between age groups (walking only). The analysis of data obtained in a sample of 215 subjects revealed minor differences in the contribution of hip and knee extensors in various synergies (as reflected by motor modules) and in the timing of synergistic muscle activity (as reflected by primitives).

By studying the effect of gender on motor control, this paper addresses an important and still understudied subject. Personally, I have never worked on muscle synergies and had to read up on the non-negative matrix factorization technique. Given my lack of experience, I am not qualified to judge the methodological procedure but understand that it represents the preferred synergy extraction method (Rabbi et al., 2020, Sci Rep). I was much impressed by the apparent efforts the authors have put into data acquisition (a sample size of 215 is way beyond the limits typically seen in basic biomedical research in humans in vivo), analysis and presentation. In this regard, I would like to sincerely thank the authors for also providing their raw data and R-code, which I found to be both instructive and very elegant.

In terms of critique, I can only mention that I found the shear amount of information presented difficult to absorb. I also expect other readers not familiar with motor control research and, specifically, techniques for the extraction of muscle synergies to have a hard time going through roughly 20 manuscript pages and a total of no less than 14(!) figures, trying to grasp the central and most important findings. With regard to figures, please also note that in some cases the legends are numbered incorrectly and do not match the respective figures. With simplicity in mind, I would have preferred if synergy results had only been reported for those muscles that are primarily responsible for the execution of a given phase of the gait cycle. This is because I am wondering whether differences in the synergy of muscle activity are physiologically relevant if the gross activity of the respective muscles is very low to start with. In figures 4 and 5, vertical lines or boxes might be introduced to highlight the prime movers of each phase. Similarly, in figure 5b the co-activations of muscles with little activity (the ones shown in semi-transparent colors) could be entirely omitted. Finally, I would have loved to see a bulleted list (maybe at the end of the Discussion) summarizing the most important sex- and age-specific differences in the degree and timing of muscle synergies.

Other than that, I would like to congratulate the authors for their beautiful work, which will certainly be of interest to the readership of PLOS ONE.

Reviewer #2: General comments:

The paper is well written and it is easy to follow the lines of argumentation. The authors address an important issue, by pointing out the lack of studies involving or focusing on female participants in clinical research. By applying the muscle synergy concept, the authors aimed at drawing conclusions about possible sex-dependent differences in muscle coordination patterns out of their analysis. An additional aim was to analyze whether these differences are maintained in older age. Thereby, the authors do not solely focus on the muscle synergy results but also take the analysis of spatio-temporal parameters into account.

To do so, the authors combined data from three different data collections (G1, G2, G3). For me, it is not clear whether these data have been published before? As far as I understood the manuscript, G1 and G2 were used to analyze the differences between females and males during walking, ones in younger age (G1) and ones in older age (G2). G3 was used to analyze the differences in running in younger age. It is mentioned that both fixed and self-selected speeds were chosen and that this information is included in the supplemental material. However, I think the information about whether the speeds mentioned in Table 1 were self-selected or prescribed should also be included in the manuscript. The motor synergies were compared using both a linear (duration of activity and point in time of main activation) and a nonlinear metric (local and global complexity of the signals). The authors choose to calculate Bayesian statistics. Unfortunately, I am not familiar with Bayesian statistics and I am thus not able to comment on this part of the manuscript.

I was not sure about some parts in the presentation of the results of the gait cycle parameters. These points are pointed out below. Four synergies have been determined, which account for 80-85% of the explained variance. Is there a specific reason that the authors did not choose the common threshold of 90%, which might have led to five synergies? Generally, the authors could consider highlighting of large effects in the figures.

There are some minor points concerning the discussion section, which are pointed out below. I am though missing a paragraph referring to the limitations of the study, e.g. the three different procedures and the various walking/running speeds.

Specific comments:

Results-section: It might be easier to follow if the results for the gait cycle parameters would be better separated in results concerning walking and results concerning running.

line 97: Figure 1 or S1?

lines 104ff: “The swing duration […] was moderately affected by both sex […] and age …” does this refer to running? If yes, this is somewhat contradictory to line 107/108: “In running, swing duration was not clearly affected by sex”

line 132: The modules can also be seen in Figure 4? Figure 4 and Figure 5 are in general somewhat redundant – can’t they be combined?

lines 133/134: I am not sure about the statement(s) that one or the other muscle contributes more or less – mustn’t we consider both the module(s) and the primitive(s) in order to make this statement?

line 148: Figure 5B?

lines 216ff: Considering the muscle-synergy-studies about walking and running, are there any having female participants? If yes: could the results/tendencies found here be linked to previous results concerning the muscle synergies in males in females (even though no direct comparisons have been made)? If no: this could be emphasized as novelty (besides the analysis of sex-specific differences).

lines 227ff: Could you please point out more specifically what you would like to say with this sentence?

line 247: Which past results do you refer to?

lines 258ff: I am sorry, but to me, the connection is not clear.

lines 261ff: Do you have a possible explanation for this contradiction?

line 275: “are not be necessarily”: I think there is a typo.

lines 307ff: Maybe you could mention which treadmill belongs to which dataset.

lines 318ff: Have you considered including adductor muscles? These might be interesting for sex-related differences (even though hard to derive).

lines 331ff: Have you evaluated the influence of the three different segmentations on a sample data set?

line 348: What is the unit of the stance and swing times?

line 356: Have you averaged the 30 cycles before applying the NMF algorithm? Earlier, you mention differences in gait cycle parameters. You normalize the stance and the swing phase separately. This can influence your results, you can lose some (timing) information - could you comment on that?

lines 376ff: Even though these methods are properly referenced, a motivation why these methods add value, since they are not commonly used (to the best of my knowledge), might be helpful for the reader. E.g., an explanation of how certain values of these metrics might be interpreted in a muscle synergy context.

lines 479 - 498: I think up from here your captions are somehow mixed up. I guess “s.d.” in Figure 5(B) signifies “standard deviation”?

Reviewer #3: 1. It is critical to calculate the sample size and power when designing the study. Based on a frequentist t-test, the sample size in G3 may only detect a medium effect size. I am not convinced that the study is sufficiently powered, especially for testing the interaction between sex and age.

2. The determination of prior information needs to be justified or citied references.

3. Did you test the interaction between speed and age, and speed and sex?

4. Age should be compared between sexes for each group to ensure the balance of age. BMI may need to compared too.

5. There are too many figures.

6. Figure 1, 6 and 8 can be simplified. Just show either boxplot or dotplot, but definite not three types altogether for each outcome. The current boxplot does not seem correct, as outliers outside 1.5* interquartile range were not shown.

7. I had a hard time to understand the figures for effect size.

The distribution figure is not necessary as almost all of the posterior distributions seem to be normal. Consider present a forest plot with effect sizes and 95% credible intervals.

Does “Sex:Age” refer to the interaction between sex and age? What does a negative/positive effect size of interaction mean?

8. It seems that none of the interactions are significant. However, it is not conclusive whether the interactions are really not significant or the study is lack of power to detect the significance.

6. PLOS authors have the option to publish the peer review history of their article (what does this mean?). If published, this will include your full peer review and any attached files.

Reviewer #1: No

Reviewer #2: No

Reviewer #3: No

---

## [Author Response · Author response to Decision Letter 0]

3 Mar 2022

Please find an extended response to the reviewers in the relevant file.

---

## [Decision Letter · Decision Letter 1]

23 May 2022

Sex-specific tuning of modular muscle activation patterns for locomotion in young and older adults

PONE-D-21-25840R1

Dear Dr. Santuz,

We’re pleased to inform you that your manuscript has been judged scientifically suitable for publication and will be formally accepted for publication once it meets all outstanding technical requirements.

Kind regards,

Peter Andreas Federolf

Academic Editor

PLOS ONE

Reviewers' comments:

Reviewer's Responses to Questions

**Comments to the Author**

1. If the authors have adequately addressed your comments raised in a previous round of review and you feel that this manuscript is now acceptable for publication, you may indicate that here to bypass the “Comments to the Author” section, enter your conflict of interest statement in the “Confidential to Editor” section, and submit your "Accept" recommendation.

Reviewer #2: All comments have been addressed

Reviewer #3: All comments have been addressed

2. Is the manuscript technically sound, and do the data support the conclusions?

Reviewer #2: Yes

Reviewer #3: (No Response)

3. Has the statistical analysis been performed appropriately and rigorously? 

Reviewer #2: I Don't Know

Reviewer #3: (No Response)

4. Have the authors made all data underlying the findings in their manuscript fully available?

Reviewer #2: Yes

Reviewer #3: (No Response)

5. Is the manuscript presented in an intelligible fashion and written in standard English?

Reviewer #2: Yes

Reviewer #3: (No Response)

6. Review Comments to the Author

Reviewer #2: I thank the authors for their time invested in reviewing the paper and their precise answers to the comments.

Reviewer #3: (No Response)

7. PLOS authors have the option to publish the peer review history of their article (what does this mean?). If published, this will include your full peer review and any attached files.

Reviewer #2: **Yes: **Felix Möhler

Reviewer #3: No

---

## [Editor Report · Acceptance letter]

24 May 2022

PONE-D-21-25840R1 

Sex-specific tuning of modular muscle activation patterns for locomotion in young and older adults 

Dear Dr. Santuz:

I'm pleased to inform you that your manuscript has been deemed suitable for publication in PLOS ONE. Congratulations! Your manuscript is now with our production department. 

Kind regards, 

on behalf of

Dr. Peter Andreas Federolf 

Academic Editor

PLOS ONE